# Simulation Analysis of Working Circuit Performance of Mountain Pepper Harvester Based on Improved Load-Sensitive System

**Di Wu \*, Zhihao Ma, Jianlong Zhang, Weiping Xu, Haifeng He and Zhenlin Li**

College of Mechanical and Electrical Engineering, Guizhou Normal University, Guiyang 550001, China;
21010200645@gznu.edu.cn (Z.M.); zhangjlztt@163.com (J.Z.); 100409206@gznu.edu.cn (W.X.);
haifenghe@gznu.edu.cn (H.H.); 222100200665@gznu.edu.cn (Z.L.)
**\*** Correspondence: 201907001@gznu.edu.cn

**Abstract:** China's Guizhou is a typical karst landscape province with high production of chili pepper, but it is mostly planted in mountainous areas, while manual harvesting of chili pepper has the deficiencies of high labor intensity, low efficiency, and high labor cost; in addition, there is no harvesting machinery applicable to the dense planting pattern of the chili pepper in mountainous areas in China. The fully hydraulic mountain track-based self-propelled pepper harvester 4JZ-1.0A is designed to solve the above problems. The pepper harvester spiral comb picking head is an important part of the whole machine design, the design of the hydraulic system of the working circuit of the picking head is the key to realizing the hydraulic control part of the whole system. In this paper, the working principle diagram of the improved load-sensitive hydraulic system is designed and analyzed for the study of whether the working circuit of the pepper picking head of the pepper machine can meet the requirements of mountain operation, taking the working circuit of the mountain pepper harvester as the research object. In addition, the load-sensitive pump model and the simulation model of the whole working circuit are established by the AMESim platform 2019.2 (Siemens simcenter amesim). The operating performance of the system under variable flow conditions, variable load conditions, and an improved sensitive system is analyzed. The simulation results show that the improved load-sensitive system can effectively reduce the oscillation and cavitation during cylinder operation and improve the system efficiency and the performance and service life of the components. The performance of the hydraulic system of the working circuit of the mountain pepper harvester was verified in the test, meeting the requirements of working use. This provides a theoretical basis for the improvement and optimal design of a mountain pepper harvester hydraulic system.

**Keywords:** mountain pepper harvester; improved load sensitivity; working circuit; AMESim simulation

## 1. Introduction

China's mountainous areas account for about 33% of the country's total land area, the hills account for about 10%, while the plains account for only 12%, and the arable land in mountainous and hilly areas accounts for a large proportion of China's arable land area [1,2]. Guizhou is a typical karst landscape province in China, 92.5% of the area is mountainous and hilly, but the Guizhou chili industry advantages are outstanding. It is estimated that, by 2022, the chili planting area will be stable at 5.3 million mu, chili production up to 6.3 million tons, an output value of 19 billion yuan, the production and sales scale as the first in the country [3]. However, the mechanical rate of major crops in Guizhou is 46%, and realizing the mechanization of pepper harvesting in Guizhou can reduce the amount of farmers' labor and increase income. Nowadays, most of the pepper harvesting machines are designed for use in the plains, but also can be oriented to the terrain in not complex mountainous areas, something Guizhou's complex terrain

environment is not able to realize. It is crucial to realize automated harvesting of pepper machinery in mountainous areas of Guizhou.

The mountain hills harvesting work state and working environment are different, unlike the plain terrain, which is flat and suitable for large mechanical movement. For example, the OXBO Axbo pepper harvester produced by Tenfold in the United States and the two-row traction pepper harvester developed by Pik Rite Company, etc. Although the power output is stable, the harvesting efficiency is high, and the light fruit contains a low miscellaneous rate, the harvester is not easy to turn, and is only applicable to long strips in a large field monopoly planting mode [4,5]. Compared to foreign countries, although the technical level of domestic agricultural machinery lags behind more, there has been rapid developments, one after another, to achieve the mechanization of the whole process of rice, corn, and other crop production, but the research on the hydraulic technology of pepper combine harvesting machinery is relatively late, until the last few years on the market there have emerged, one after another, fully hydraulic types of pepper harvester. Hebei Rei Ken Agricultural Machinery Co., Ltd. (Anlu, China) produced and listed the 4JZ-3600 pepper harvester based on 4AZ-2200 in 2018, the maximum power of this model is 62 kw, the net harvesting rate is up to more than 90%, the rate of impurity is less than 10%, the total loss rate is no more than 6%, and the operating hour is up to 0.6 hectares. Xinjiang Mu Zhen in 2021 introduced the new Mu Zhen 4JZ-3600B with an unloading height of up to 4.1 m, reducing the number of stops to unload and the unloading time, improving the transport efficiency of the field and the productivity of the machine [6,7]. The mountain pepper harvester will work with constant climbing and downhill operations and variable working conditions, which requires the working circuit of the pepper harvester to work continuously and steadily under complex working conditions, i.e., without the influence of the traveling circuit, so large domestic and foreign pepper harvesters are not suitable for mountainous and hilly terrain.

Load-sensitive technology has been applied to mountain pepper harvesting in order to cope with the complex working environment of mountain pepper harvesters and to reduce the energy consumption of harvesters. Load-sensitive systems were first used in construction machines with internal combustion engines, such as excavators and non-excavating drills. In the 1980s, energy conservation became the focus of researchers, and De-Zhong et al. addressed the problems of temperature rise and poor control performance of conventional coal drilling rigs by making load-sensitive hydraulic systems available for coal drilling rigs to improve performance [8]. Tian-Liang Lin et al. proposed a load sensing system for electric excavators based on displacement adaptive variable speed control, and the system and control strategy can effectively improve control performance and reduce energy consumption by 19–70% [9]. Shen Hui Jun et al. proposed adding energy-saving control valves to reduce the load-sensitive hydraulic system with the differential compensation pressure of loader multiway valves, which reduces the energy loss of multiway valves and improves the performance and service life of the system and components [10]. In response to the problems of the hydraulic system of the roadhead machine, such as quantitative pump supply, neutral function of reversing valve to unload, high energy consumption of the system, and complicated design of the reversing main valve spool, Q P designed the load-sensitive system of rocker lift of the roadhead machine [11]. With the development of agricultural machinery, load-sensitive technology was rapidly applied to agricultural machinery. Luo Y L et al. have applied a load-sensitive technology to agricultural machinery, such as mountain mowers, rototillers, and mountain tractors, many times to make agricultural machinery adapt to complex and harsh working conditions, which are well divided for state control in order to match the power system with the load power and avoid engine power loss [12–14]. Chen Y L et al. applied the load-sensitive technology to the hydraulic system of the cutting roller of a sugarcane combine harvester, and the energy consumption and efficiency of the load-sensitive system were 52% and 1.85 times the throttle speed regulation system, respectively, with remarkable energy-saving effect [15]. Chen Jincheng et al. designed the spray bar suspension hydraulic system based on load-sensitive theory

and load theory; the system pressure is always compatible with the load pressure, and the system energy consumption is low during the operation of the hydraulic system [16]. Load-sensitive systems are widely used in fields such as construction and agricultural machinery for their economy, proportionality, and advancement.

In this paper, we propose a load-sensitive hydraulic system for a mountain pepper harvester with improved valve front compensation. The load-sensitive system with improved valve front compensation adds a counterbalance valve and a reducer valve for charge oil to improve the working performance. The results show that the performance of the designed working circuit hydraulic system meets the requirements of use, which is a good guideline for the design and selection of the hydraulic system of the mountain pepper harvester.

## 2. Working Circuit Hydraulic System Design and Working Principle Analysis

The shape of the mountain pepper harvester is shown in Figure 1. The working circuit of the mountain pepper harvester mainly includes two actions: the lifting of the spiral comb-picking mechanism and the rotation of the comb motor. The working performance of the mountain pepper harvester requires that these two actions can be carried out simultaneously and without interfering with each other. When performing compound actions, good speed regulation of the hydraulic system is required to meet the flow and pressure required for each action, respectively [17]. In load-sensitive systems, post-valve compensation only ensures that the differential pressure across the valve ports is equal, and the differential pressure across the valve ports is not a constant, requiring a differential pump pressure compensator to coordinate to improve speed regulation rigidity, so the system has poor speed regulation rigidity. While the compensation before the valve directly sets the compensator, the differential pressure can more accurately control the speed of the load, especially with each valve port using an independent compensator, the system speed regulation performance is better [18]. For the stable working environment of the hydraulic motor, this paper weakens the influence of the system on the motor, while for the same working conditions and role of the double lift cylinder, the double lift cylinder is simplified. In this study, a set of the mountain pepper harvester working circuits is designed according to the principle of pressure compensation before the valve, and its schematic diagram is shown in Figure 2.

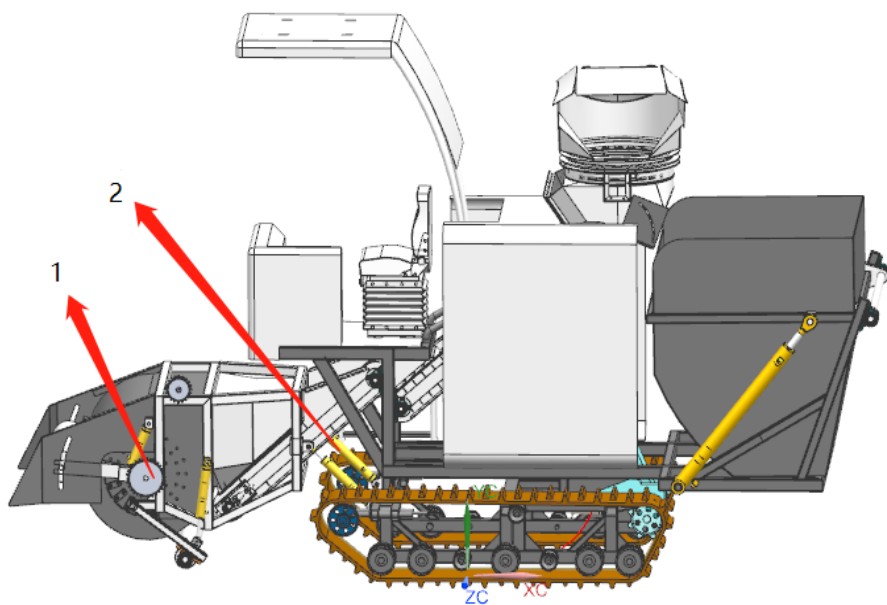

**Figure 1.** Schematic diagram of the mountain pepper harvester. 1 Comb rotating motor; 2 screw comb picking mechanism lifting cylinder.

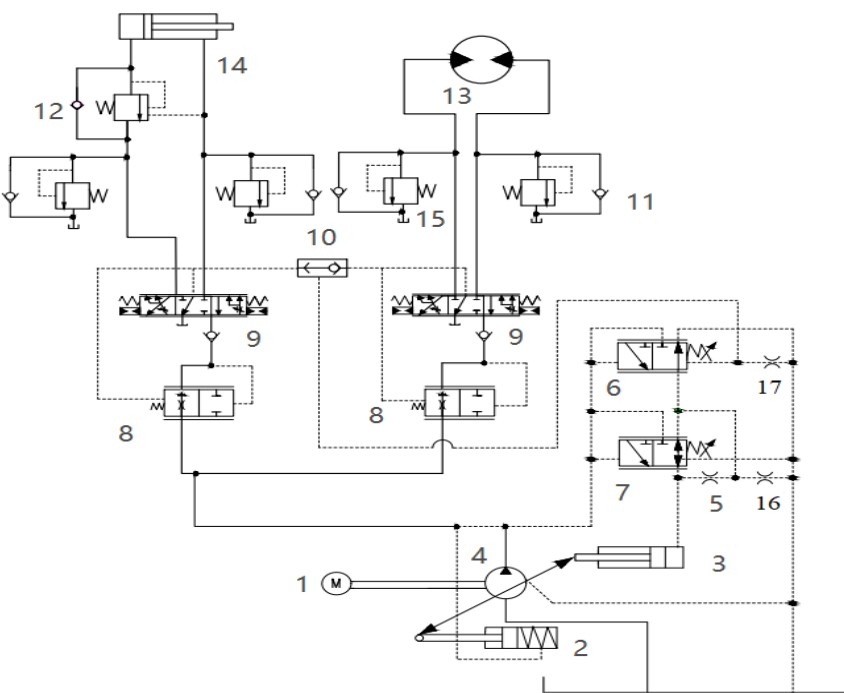

**Figure 2.** Principle diagram of the working circuit of mountain pepper harvester. 1 Motor main drive; 2 variable cylinder spring chamber; 3 variable piston cylinder; 4 variable pump; 5, 16, 17 damping holes; 6 load sensitive valve; 7 pressure compensation valve; 8 pressure compensator; 9 load sensitive multiway valve; 10 shuttle valve; 11 charge pressure reducing valve; 12 balance valve; 13 comb rotary hydraulic motor; 14 lift cylinder; 15 oil tank.

The load-sensitive control system of the working circuit of the mountain pepper machine is analyzed by means of Figure 2. The load-sensitive pump consists of motor main drive1, variable cylinder spring chamber2, variable piston cylinder3, variable pump4, damping bore5, load-sensitive valve 6, and pressure compensation valve7. A portion of the oil loaded in the system will return to the tank through damping holes 16 and 17. Although some efficiency will be lost, this will reduce pressure shocks during the movement of the spool of the load-sensitive and pressure-compensating valves, preventing pressure blockage in the closed center of the multiway valve system [19]. The control oil leading from the pump through the valve port of the load-sensitive valve (can be regarded as variable liquid resistance) and damping hole 17 can form a C-type hydraulic half-bridge; when the left position of the pressure compensation valve works, the pressure compensation valve port and damping holes 5, 16 in series with the fixed liquid resistance of the C-type hydraulic half-bridge and the two C-type hydraulic semi-variable piston cylinder fluid pressures, the change of the valve port will not cause the variable piston cylinder in the oil pressure to undergo drastic change, can improve the control stability, pump displacement control stability, but also help to reduce the pressure impact on the actuator, in the structure of a reasonable arrangement of damping holes, this set-up can improve the dynamic response performance of the system, improving the response stability of the system [20]. The pressure compensator 8 is used to maintain the differential pressure between the front and rear of the multiway valve 9, so that the differential pressure between the front and rear of the multiway valve remains constant, and the multiway valves each lead one way to the shuttle valve 10 for load-sensitive control [21]. When the required flow rate of the load is larger than the pump flow rate, the load-sensitive valve is in the left position, the pump is in the maximum displacement, the difference between the pump pressure and the load pressure gradually becomes smaller, and when the difference is smaller than the value set by the compensation valve, the compensator opens to provide the flow rate for the small load [22]. When the flow requirements of the small load are met, then the flow is

supplied for other loads to better meet the flow requirements of different loads and meet the requirements of mountain pepper harvesting.

## 3. Modeling of the Hydraulic System of Mountain Pepper Harvester

To check whether the designed working circuit meets the working requirements, the working circuit is modeled in AMESim [23]. For relatively complex components modeled using the HCD library, the built load-sensitive pump and working circuit models are shown in Figure 3. Simulation and analysis of the models are performed to verify the rationality and reliability of the hydraulic system of the working circuit.

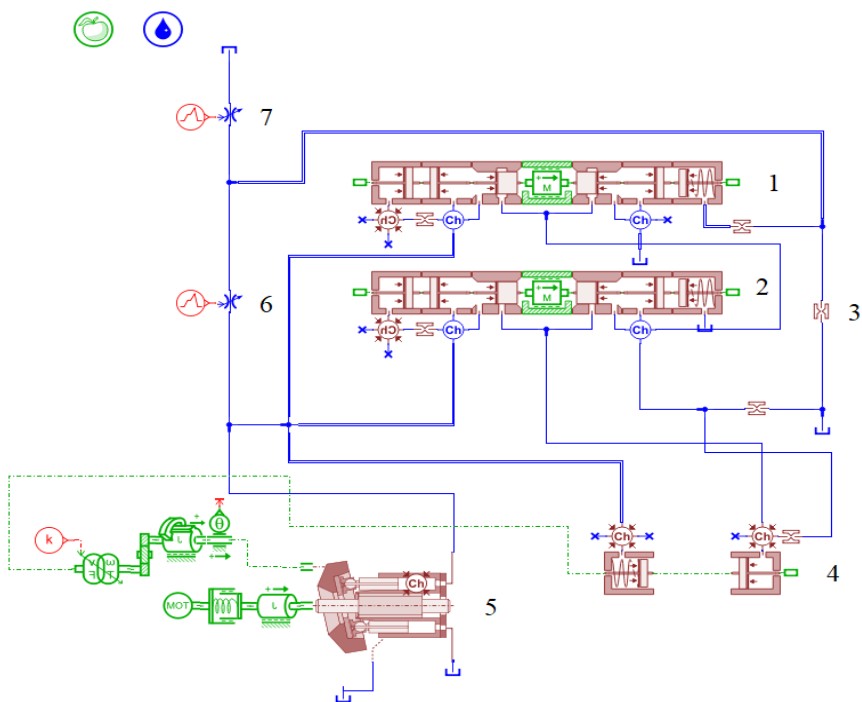

**Figure 3.** Load-sensitive pump verification loop. 1 LS valve; 2 pressure compensation valve; 3 throttle orifice; 4 variable cylinder; 5 variable piston pump; 6 analog reversing valve; 7 variable load.

### 3.1. Load-Sensitive Pump Model Construction

Due to the complexity of load-sensitive pumps, a load-sensitive pump circuit model was built in AMESim, as shown in Figure 3, for study and analysis [24]. The variable damping orifice is controlled by a gradually decreasing control signal from 0 to 10 s, which is used to simulate the external load of the load-sensitive pump variation. The simulated reversing valve 6 is kept open to the maximum, and the operating state of the load-sensitive valve and pump is analyzed to verify whether the load-sensitive circuit meets the operating performance requirements.

From Figure 4, it can be obtained that the simulated reversing valve 6 shows pressure fluctuation at the start-up of the system from 0 to 0.3 s, and the pressure difference before and after the valve is always maintained at 14 bar from 0.3 to 7.6 s. The load-sensitive pump continuously adjusts the pump outlet pressure from 0.3 to 7.6 s, and the pump pressure reaches the maximum value of 240 bar after 7.6 s and remains constant. From Figure 5, it can be analyzed that the opening of the load-sensitive valve is maximized from 0.3 to 7.6 s to control the oil circuit and the pressure compensation valve does not work. A total of 7.6 s later, the opening of the load-sensitive valve gradually decreases, and the opening of the pressure compensation valve increases to protect the oil circuit and reduce the pump outlet discharge. As a result, the load-sensitive circuit can maintain a stable flow rate when executing external load, which verifies the effectiveness of the load-sensitive circuit.

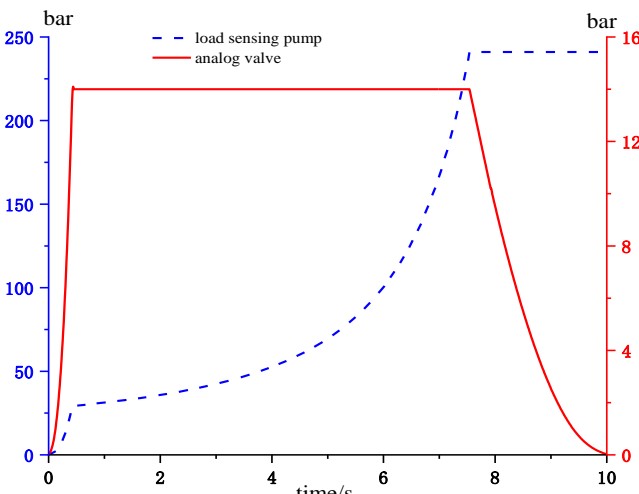

**Figure 4.** Differential pressure before and after the valve and load-sensitive pump pressure curve.

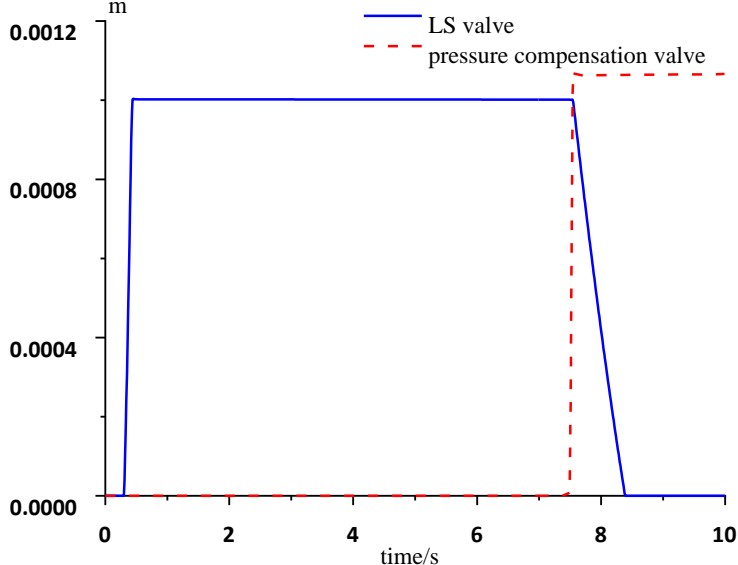

**Figure 5.** Load-sensitive valve and pressure-compensated valve spool displacement curve.

### 3.2. Working Circuit Model Building

According to the improved load-sensitive hydraulic system, a simulation model of the working circuit system of the mountain pepper harvester is established using AMESim software, as shown in Figure 6. According to the structure and working principle of load-sensitive hydraulic system components, the parameters of each main module of AMESim are set as shown in Table 1, and other parameters are kept as default.

**Table 1.** Main data of working circuit simulation model of mountain pepper harvester.

| Main Parameters | Numerical Value | Main Parameters | Numerical Value |
|---|---|---|---|
| Hydraulic cylinder bore/mm | 80 | Pressure compensation valve spring force/N | 660 |
| Hydraulic cylinder rod diameter/mm | 40 | Multiway valve spool displacement/m | 0.006 |
| Hydraulic Pump Displacement/mL·r$^{-1}$ | 250 | Balancing valve setting pressure/bar | 200 |
| Hydraulic pump motor speed/r·min$^{-1}$ | 1500 | Multi-way valve spool diameter/mm | 10 |
| Load sensitive valve spring force/N | 40 | Rated speed of hydraulic motor/r·min$^{-1}$ | 258 |

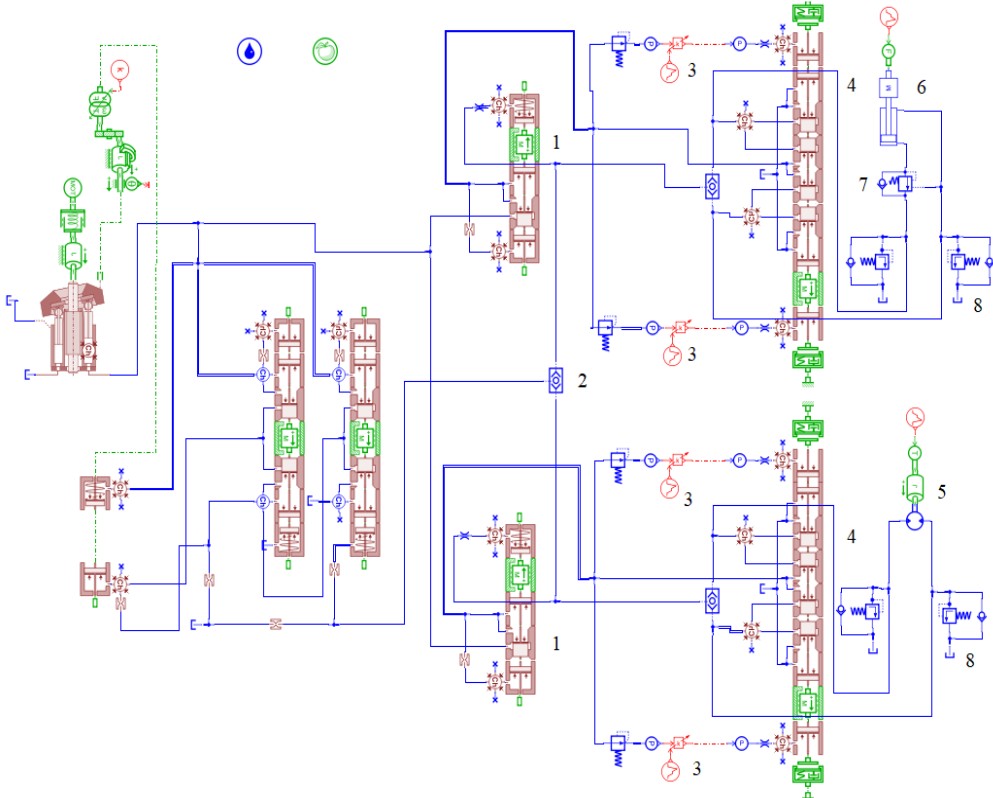

**Figure 6.** Working circuit simulation loop model. 1 Pressure compensator; 2 shuttle valve; 3 multiway valve control signal; 4 multi-way valve; 5 comb rotary motor; 6 lift cylinder; 7 balancing valve; 8 relief valve.

## 4. Characterization of the Hydraulic System of Mountain Pepper Harvester

### 4.1. Variable Flow Rate System Analysis

When the mountain pepper harvester was harvested stably on a level road, the external load of the working circuit was basically constant, setting a constant external load of 500 N for the hydraulic cylinder and a constant torque load of 50 Nm for the hydraulic motor. The flow resistance of the system and pressure variation were examined under a constant external load and flow variation at different opening areas of the two multiway valves, and the pressure variation curves of the load-sensitive system are shown in Figures 7 and 8 [25].

From the analysis of Figure 7, the pressure difference between the front and rear of the multiway valve is stable at about 7 bar, the pressure difference between the front and rear of the valve is equal to the pressure before the valve minus the compensator spring force; when the spring force becomes larger, the corresponding pressure compensation decreases, so setting a reasonable spring force affects the pressure difference between the front and rear of the multiway valve. The compensator sets the spring force at 40 N, and the pressure difference between the LS valve and the pump outlet is stabilized at about 22 bar. In 5.2~5.9 s, the pressure difference is about 0 bar, which means that the LS valve performs pressure cut-off, the load-sensitive system does not work, the pump opening is maximum, there is maximum flow rate of the given system, and the resistance to flow saturation is strong. The comparison of the pressure before and after the multiway valve when the multiway valve is fully open and the comparison of pressure between the pump outlet and LS valve are shown in Figure 8. The pressure difference before and after the valve is basically unchanged, and the pump flow is fully opened in 5.2~6.8 s and then controlled by the LS valve. By comparing Figures 7 and 8, it can be obtained that the size of the valve opening, i.e., the size of the valve flow, does not affect the normal movement of the mechanism, and the oscillation time of the system increases by 0.6 s at high flow rates,

and the maximum opening time of the pump increases by 0.9 s, which has a small impact on the system, with good anti-saturation capability.

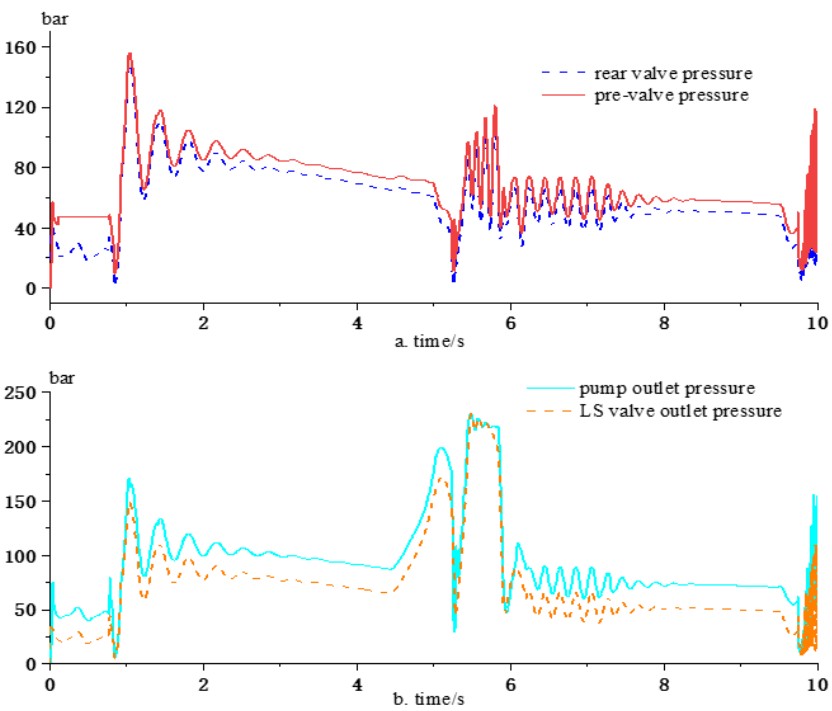

**Figure 7.** Half-open pressure comparison curve of multiway valve port. (**a**) Pressure comparison before and after multiway valve; (**b**) pressure comparison between pump outlet and LS valve.

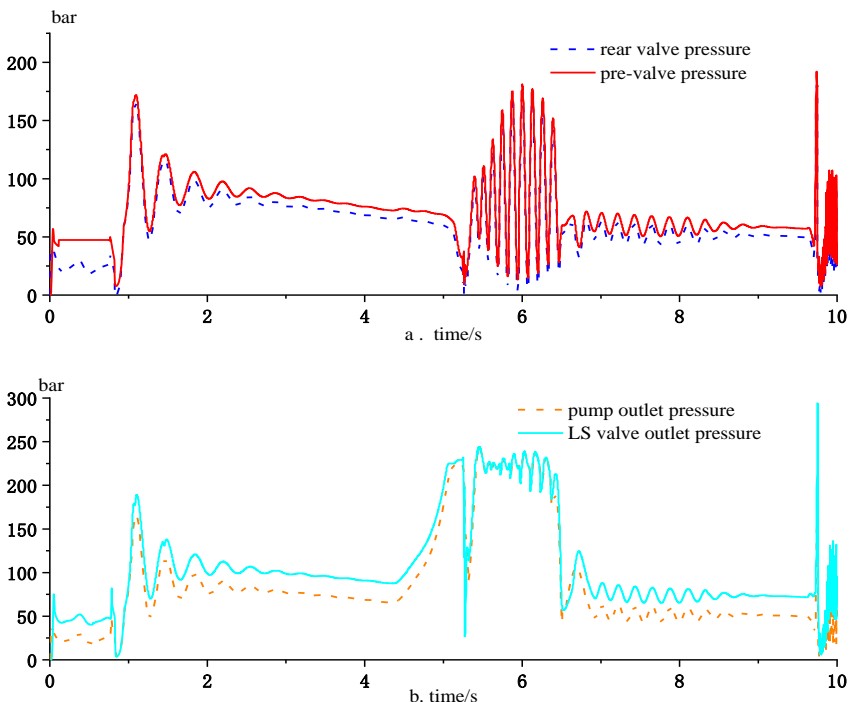

**Figure 8.** Multi-way valve port fully open pressure comparison curve. (**a**) Multi-way valve before and after pressure comparison; (**b**) pump outlet and LS valve pressure comparison.

### 4.2. Characterization of Variable Load Conditions

The working scenes of the mountain pepper harvester are mostly mountainous and have a hilly terrain. In order to simulate the change of external load brought about by the



terrain, we set the external load signal of the hydraulic cylinder in Figure 6 to gradually increase the load from 0 N to 500 N within 10 s; the external load torque of the hydraulic motor also increases from 0 to 50 Nm within 10 s, thus imitating the load change working condition. Given the multiway valve signal as shown in Figure 9a, the right side of the multiway valve is powered within 0~5 s, the left side of the multiway valve is powered within 5~10 s, and the multiway valve works at full flow rate.

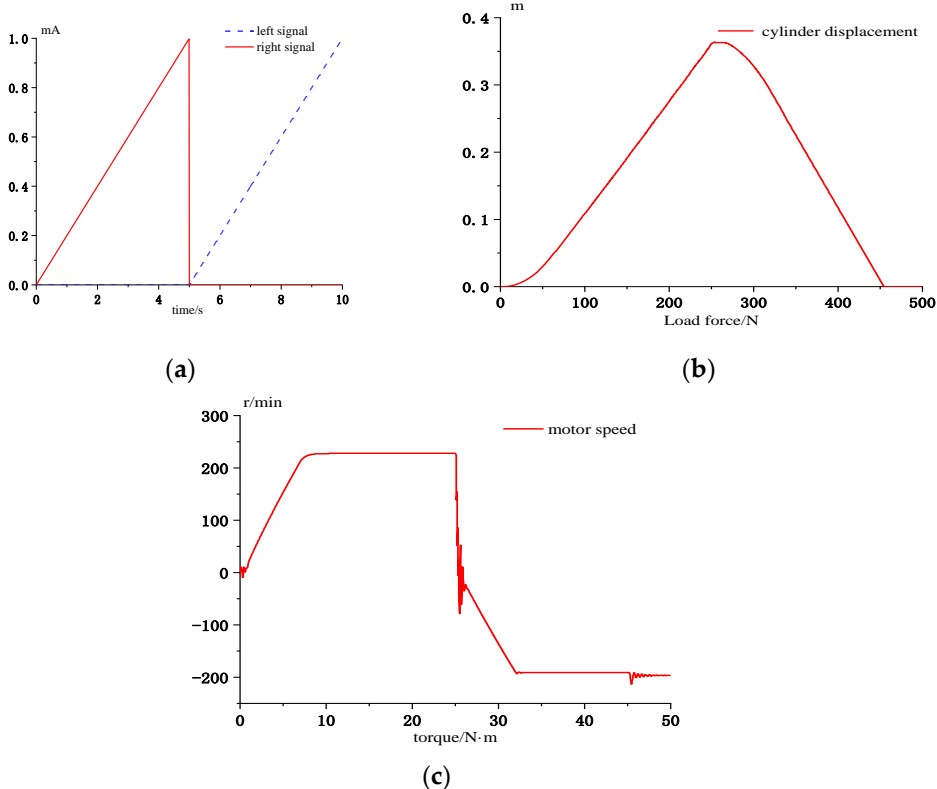

**Figure 9.** (**a**) Signal curve to the left and right of a given multiway valve; (**b**) lifting cylinder displacement and external load curve; (**c**) comb rotary motor speed vs. external load curve.

From the analysis of Figure 9b, it can be seen that in the first 0.8 s of response time the cylinder speed is slow; after that, the cylinder continues to elongate, the speed becomes faster, in the 5 s the multicircuit valve reverses, the cylinder began to decline, but with the increase of the external load, the cylinder descending speed becomes faster, which indicates that the external load has a certain effect on the speed of the cylinder, but the speed of the cylinder is smooth, and it can cope well with the impact of external loads, and meets the spiral comb-type Picking mechanism lifting mechanism requirements. Analyzed by Figure 9c, there is a certain pressure fluctuation when the motor starts and when it turns, but basically it is not affected by the external torque, and the maximum speed is stable at 238 r/min, and it is verified by the test that, when the speed of the comb rotary motor is at 230 r/min, the harvesting of chili peppers has the best effect of the picking rate and the breakage rate, which is in line with the expected effect [26]. The analysis shows that the actuator motion has good adaptability to different working environments and is less affected by external loads. The control characteristics of the working circuit of the mountain pepper harvester are good, the low-load circuit in the system is not affected by the high-load circuit, and each circuit can work alone.

### 4.3. Improved Load-Sensitive System Characterization

4.3.1. Characterization after Adding the Charge Reduction Valve

In the hydraulic system, cavitation and pressure shock may occur, which may damage the inner wall of the cylinder and cause leakage, affecting the service life. From Figure 10a,

the pressure curve of the cylinder's rodless cavity appears negative in 8.7~9.6 s without adding the oil replenishment pressure-reducing valve, which indicates that the oil cannot meet the fluid supply demand when the cylinder is in motion, resulting in oil cavitation and cavitation of the cylinder, which corrodes the cylinder and reduces the accuracy and service life of the cylinder. After adding the fill pressure-reducing valve, the oil can be filled directly from the tank through the one-way fill valve. It can be seen visually in Figure 10a that, after adding the valve, the oil absorption phenomenon is basically eliminated, as expected. From Figure 10b, the pressure fluctuation in the rod cavity is large in 8.5~9.7 s, and there is a high-pressure wave, which will have a large impact on the cylinder, and this can easily cause wear at the cylinder's inner wall and cause oil leakage. The pressure-reducing valve is set to 200 bar, and the gradient of flow capacity is 20 L/min/bar. After adding the pressure-reducing valve, the pressure fluctuation is obviously reduced, and the pressure is stable at 200 bar within 8.5~9.7 s, which reduces the pressure shock in the rod chamber and effectively extends the service life of the components, which meets the expected effect.

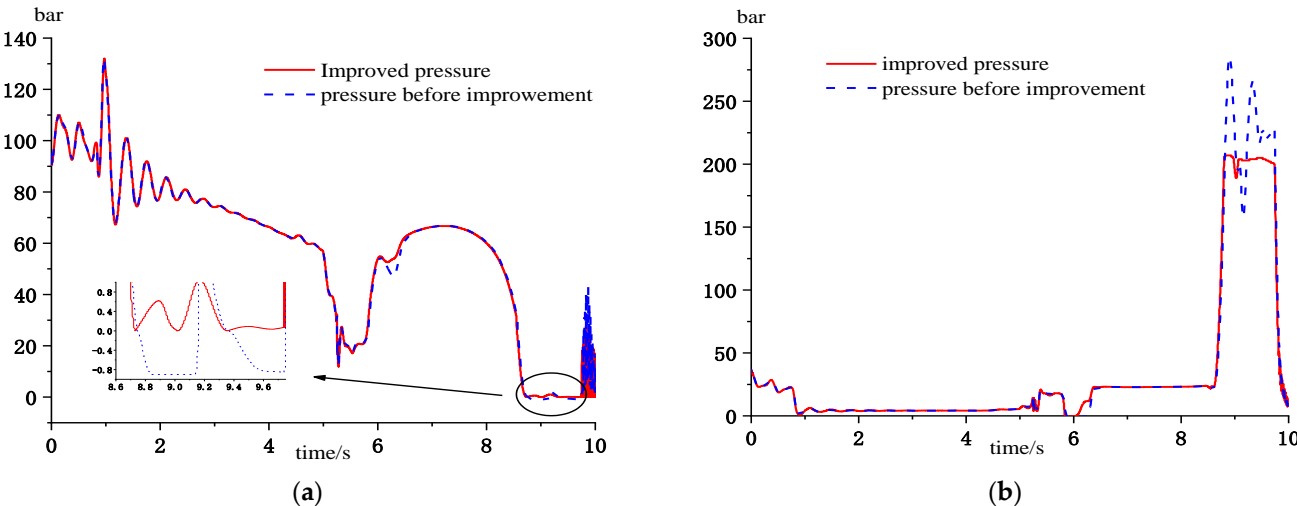

**Figure 10.** Pressure curve. (**a**) Pressure curve of cylinder rodless chamber before and after adding the charge pressure reducing valve; (**b**) pressure curve of cylinder rod cavity before and after adding the charge pressure reducing valve.

### 4.3.2. Characterization after Adding Balancing Valve

From Figure 11 it can be seen that there is no balancing valve when the cylinder speed is fast, the speed fluctuation is large, the cylinder operation is unstable, increasing the difficulty of the operation of the mountain pepper harvester; stability and operability are reduced. When the speed increases sharply, the cylinder pressure also becomes large sharply at this time. When the balancing valve is added, the pressure threshold is set at 220 bar. At this time, the oil goes directly to the small pressure chamber through the check valve to reduce the pressure shock and balance the pressure in the cylinder chamber, so the balancing valve is added to play the role of pressure stabilization and anti-vibration [27,28]. From Figure 11, it can be analyzed that the speed oscillation is not obvious in the first 5.8 s, and it has little effect on the system before and after adding the balancing valve. After 5.8 s, when the balancing valve is not added, the system appears to have high-speed movement, the movement speed is unstable, and the speed changes drastically, but after adding the balancing valve, the speed fluctuation decreases by 52% and stabilizes at a certain speed, and the system speed only produces small fluctuations [29]. After adding the balancing valve, the speed fluctuation and oscillation of the harvester become smaller, the maneuverability and comfort of the harvester is better, the operation speed is more stable, and the safety of the harvester is improved [30].

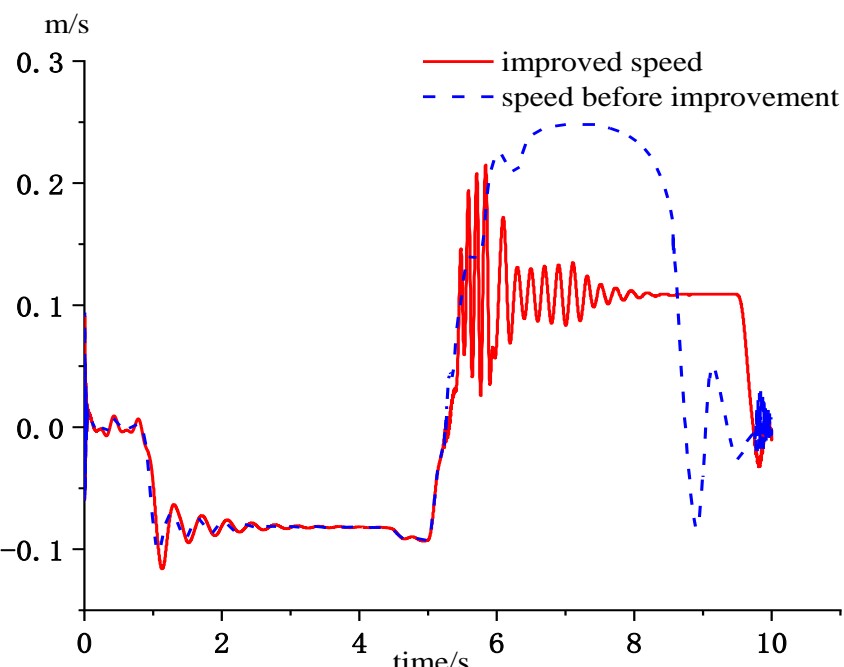

**Figure 11.** The speed curve of the cylinder before and after adding the balance valve.

## 5. Experimental Results and Discussion

### 5.1. Verification Test of the Performance of the Working Circuit System

This paper carries out simulated harvesting tests on a mountain pepper harvester, using test equipment to test the actuating components such as comb hydraulic motors and lifting hydraulic cylinders for the harvesting requirements to be met and to study the dynamic characteristics of the hydraulic system.

#### 5.1.1. Lift Cylinder Performance Test

Test the performance of the lifting cylinder: manipulate the picking head from the ground to the highest position, and then from the highest down to the ground (Figure 12b); this action was repeated five times. At the same time, the time required for the lift cylinder to rise and fall and the displacement value of the cylinder piston rod at the highest position of the picking head were recorded. Finally, the average value was calculated as the expansion and contraction speed of the cylinder. When the picking head was lifted to the highest position, then the engine was turned off and left for 30 min, the displacement of the cylinder piston rod was measured again, and the displacement deviation was calculated twice as the displacement deviation of the cylinder and the deviation rate was calculated by measuring five times, and the test results of the lifting cylinder were shown in Table 2.

**Table 2.** Lift cylinder test results.

| Number of Times | Cylinder Displacement (mm) | Boost Time (s) | Boost Speed (m/s) | Descent Time (s) | Reach Back Speed (m/s) | Displacement Deviation Amount (mm) |
|---|---|---|---|---|---|---|
| 1 | 365.5 | 4.9 | 0.075 | 4.0 | 0.093 | 1.9 |
| 2 | 368.2 | 4.7 | 0.079 | 3.8 | 0.096 | 2.1 |
| 3 | 364.3 | 4.8 | 0.076 | 3.7 | 0.095 | 1.6 |
| 4 | 368.3 | 5.0 | 0.074 | 3.9 | 0.094 | 1.3 |
| 5 | 369.8 | 4.7 | 0.078 | 4.0 | 0.092 | 1.8 |

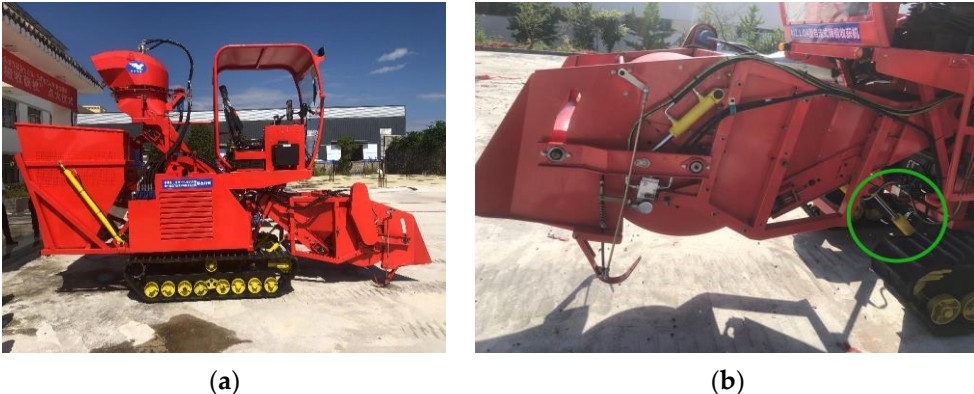

|  (a)  |  (b)  |

**Figure 12.** Physical Prototype Testing. (**a**) Physical prototype of mountain pepper harvester; (**b**) prototype lift test chart.

5.1.2. Comb Motor Performance Test

For the comb motor, the speed sensor and torque sensor were installed on the motor output shaft. The brake was installed on the other side to simulate the load torque. The motor speed was observed by adjusting the tension of the brake to change the motor load, and the speed test results of the comb hydraulic motor were recorded five times continuously, as shown in Table 3.

**Table 3.** Comb hydraulic motor speed test results.

| Number of Times | Load Torque (N.m) | Rotational Speed (r/min) |
|---|---|---|
| 1 | 30 | 236 |
| 2 | 35 | 230 |
| 3 | 40 | 233 |
| 4 | 45 | 234 |
| 5 | 50 | 232 |

*5.2. Experimental Results and Analysis*

The measured parameters of the lifting cylinder and motor are obtained through experiments (Figure 13), and the deviation rate between the measured parameters and the theoretical parameters is calculated to reflect the rationality of the design based on the deviation rate, which is calculated by the following formula:

$$\zeta = \frac{S - S_C}{S} \times 100\%$$

where $S$ is the theoretical value and $S_C$ is the actual value.

The analysis of the data in Tables 2 and 3 shows that the average time taken for the extension of the cutting table lifting cylinder to the predetermined position is 4.8 s, the average lifting speed is 0.076 m/s, the average reaching back speed is 0.094 m/s when the cutting table lifting cylinder is contracted from the predetermined position to the starting position, the average telescopic displacement of the cylinder is 367.6 mm, the average displacement deviation is 2.1 mm, and the average speed of the comb motor is at 233 r/min. The average deviation of the displacement of the piston rod extension and retraction of the lift cylinder from the simulation result is 0.64%, the average deviation of the piston rod extension speed from the simulation result is 0.78%, the deviation of the retraction speed from the simulation result is 4.44%, the average deviation of the comb motor speed is 2.1%, and the deviation rate of the five tests is basically stable. From the deviation calculation results, the cylinder displacement deviation is small, and the piston rod retraction speed deviation is large, indicating that when the comb picking mechanism is falling, affected by the gravity of the picking mechanism itself, the cylinder pressure

becomes large, the load-sensitive system adjusts, the cylinder operating speed is affected, but from the actual operating effect the system fully meets the operational requirements. The external environment to which the comb motor is subjected is variable, the factors encountered during the tests are variable, and the theoretical calculations and simulations are performed under ideal conditions, but these deviations are within the normal range and do not affect the operational results.

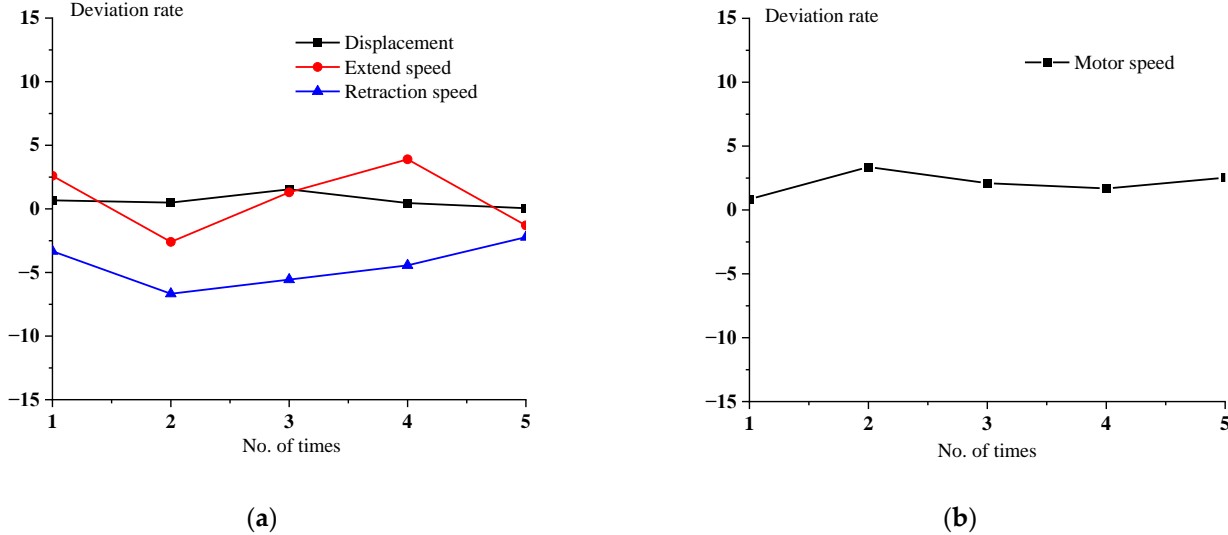

(**a**)                                (**b**)

**Figure 13.** (**a**) Lift cylinder deviation rate; (**b**) comb motor speed deviation rate.

In the actual situation, the lineup of the pipeline is not the ideal linear distribution, all of them will have some bends. The system loading start-up takes longer than the theoretical simulation, resulting in a greater deviation rate of the cylinders. The deviation rate of the circuit may be reduced if it is in continuous operation. However, during the whole test, the harvester operated sensitively and without lagging, and there was no obvious vibration impact on the hydraulic cylinder. The comb motor has a small change in rotational speed when affected by external loads, showing good working performance of the improved load-sensitive system, which meets the performance requirements of the actual work of the pepper harvester.

## 6. Conclusions

In this paper, an improved load-sensitive based mountain pepper harvester working circuit is designed, and a simulation model is built by using AMESim software, and the simulation experiment results prove that the hydraulic system is reliable. It provides a theoretical basis for all types of mountain harvester working circuit designs based on a load sensitive technology.

The paper analyzes the different output characteristics of the hydraulic system of the harvester under a variable flow condition and a variable load condition. The results show that each actuator is not affected by the external load and flow saturation, and the stability characteristics of the system are excellent. For the shortcomings, such as cavitation and oscillation generated during the operation of cylinders in the traditional pre-valve compensated load-sensitive hydraulic system, the improved load-sensitive system in this paper incorporates a make-up pressure-reducing valve and a counterbalance valve. Simulation results show that the improved load-sensitive system significantly reduces the cylinder cavitation, pressure shock and speed of the unsteady problem, and improves the performance and life of the cylinder. In order to study the effect of the improved load-sensitive system on the cylinder in the working circuit, test verification was carried out on the lifting cylinder and comb motor. The test results are in good agreement with the simulation results and meet the performance requirements of the actual work. In this

paper, the simplification of the double lifting cylinders, ignoring the impact of synchronous cylinder accuracy, and the performance of other parameters of the system (such as flow, pressure) have not been tested, so in the subsequent research and testing need to continue to explore and analyze.

**Author Contributions:** Data curation, Z.L.; Writing—original draft, J.Z.; Writing—review & editing, D.W., Z.M. and W.X.; Visualization, H.H.; Funding acquisition, D.W. All authors have read and agreed to the published version of the manuscript.

**Funding:** This work is financially supported by Guizhou Provincial Natural Science Research Program (Natural Science) (Grants No. QJH[2022]171) and Guizhou Normal University Natural Science Research Program (Grants No. QSXM[2021]B18).

**Institutional Review Board Statement:** Not applicable.

**Informed Consent Statement:** Not applicable.

**Data Availability Statement:** Not applicable.

**Conflicts of Interest:** The authors declare no conflict of interest.

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
