# Peer review of "Simulation Analysis of Working Circuit Performance of Mountain Pepper Harvester Based on Improved Load-Sensitive System"

_applsci, doi:10.3390/app131810008_

Round 1

Reviewer 1 Report

The present work is concerned with a developed chili pepper harvesting machine suitable for mountaineous Guizhou/China region. The fully hydraulic mountain track-based self-propelled pepper harvester 4JZ-1.0A designed for the above purpose is tested for labor intensity, efficiency, and labor cost. The load-sensitive pump model and the simulation model of the whole working circuit are established by the AMESim platform, and. the operating performance of the system under variable flow conditions, variable load conditions and improved sensitive system is analyzed. The improved load-sensitive system is found to effectively reduce the oscillation and cavitation during cylinder operation and improve the system efficiency and the performance and service life of the components.

The work seems to be publishable in MDPI: Applied Sciences if a careful attention is given to the following coments;

A)- There are some abbreviations and unexplained symbols used in the Abstract. Also, state that China is the country under the study.

B)- Language is fine overall, take care of typos.

C)- The literature survey is up to date and covers relevant works.

D)- In the Abstract the final sentence is imcomplete.

E)- It appears that there are other developed machines, so clearly explain why they are not fit for dense harvesting pattern of chili pepper in the mentioned area.

F)- The left and right axes of figure 4 imply pressures at different bar scales, is this true?

G)- Correct the label “pressuure compensation valve” in figure 5.

H)- Figure 9 designates negative values after some crtical torque, can they be physical?

K)- Figure 13 can be improved.

L)- The sentence “In the actual test, because the actual …” is confusing.

M)- Numerate the deviation rate formula.

N)- It is unclear how the mountain pepper harvester stability depends on the heating conditions while operating under different loads. Potential heating problems and cooling cases can be studied in a future work based on energy equation to make the created hydraulic system more reliable. Authors may mark this in their conclusion with potential references including “DOI: 10.1016/J.APPLTHERMALENG.2015.12.027” and “DOI: 10.1016/J.ENERGY.2020.117837”.

Adequate

Author Response

An example can be found here.

Reviewer 2 Report

The manuscript entitled "Simulation analysis of working circuit performance of mountain pepper harvester based on improved load-sensitive system" has been prepared by the authors. It is an applied and good topic however, it is not enough to consider in a scientific journal. In other words, it lacks innovation.

Dear Editor;

Moderate editing of English language required.

Reviewer 3 Report

The paper explain a detailed method for an agricultural machine useful for a particular application. The authors explain clearly the control design based on particular tools, The paper is report the basus for a real implementation of the proposed system

The author have to explain better the clearly improvements respect other similar systems.

Moreover they must diacuss on the possibility of using the similar approach to other machines.

The conclusions are too much schematics.

Avoid to use number for the various remarks and propose a summary discussion.

Round 2

Reviewer 2 Report

The reviewer's concerns remain.

Minor editing of English language required